# Longitudinal study of use and cost of subacromial decompression surgery: the need for effective evaluation of surgical procedures to prevent overtreatment and wasted resources

Tim Jones, [1,2] Andrew J Carr,[3] David Beard,[3] Myles-Jay Linton,[1,2] Leila Rooshenas,[2] Jenny Donovan,[1,2] William Hollingworth[1,2]

[1]The National Institute for Health Research Collaboration for Leadership in Applied Health Research and Care West (NIHR CLAHRC West), University Hospitals Bristol NHS Foundation Trust, Bristol, UK
[2]Population Health Sciences, University of Bristol, Bristol, UK
[3]Nuffield Department of Orthopaedics, Rheumatology and Musculoskeletal Sciences, University of Oxford, Botany Research Centre, Oxford, UK

**Correspondence to**
Dr Tim Jones;
Timothy.Jones@bristol.ac.uk

## ABSTRACT

**Objectives** To illustrate the need for better evaluation of surgical procedures, we investigated the use and cost of subacromial decompression in England over the last decade compared with other countries and explored how this related to the conduct and outcomes of randomised, placebo-controlled clinical trials.

**Design** Longitudinal observational study using Hospital Episode Statistics linked to Payment by Results tariffs in England, 2007/2008 to 2016/2017.

**Setting** Hospital care in England; Finland; New York State, USA; Florida State, USA and Western Australia.

**Participants** Patients with subacromial shoulder pain.

**Interventions** Subacromial decompression.

**Main outcome measures** National procedure rates, costs and variation between clinical commissioning groups in England.

**Results** Without robust clinical evidence, the use of subacromial decompression in England increased by 91% from 15 112 procedures (30 per 100 000 population) in 2007/2008, to 28 802 procedures (52 per 100 000 population) in 2016/2017, costing over £125 million per year. Rates of use of subacromial decompression are even higher internationally: Finland (131 per 100 000 in 2011), Florida State (130 per 100 000 in 2007), Western Australia (115 per 100 000 in 2013) and New York State (102 per 100 000 in 2006). Two randomised placebo-controlled trials have recently (2018) shown the procedure to be no more effective than placebo or conservative approaches. Health systems appear unable to avoid the rapid widespread use of procedures of unknown effectiveness, and methods for ceasing ineffective treatments are underdeveloped.

**Conclusions** Without good evidence, nearly 30 000 subacromial decompression procedures have been commissioned each year in England, costing over £1 billion since 2007/2008. Even higher rates of procedures are carried out in countries with less regulated health systems. High quality randomised trials need to be initiated before widespread adoption of promising operative procedures to avoid overtreatment and wasted resources, and methods to prevent or desist the use of ineffective procedures need to be expedited.

## Strengths and limitations of this study

► Our study used a national, longitudinal data set over a 10 year period covering all National Health Service (NHS) secondary care providers in England, and private provision for NHS-funded patients.

► Hospital Episode Statistics are linked to hospital payments, which is a strong incentive to provide complete data, and allowed us to explore costs of subacromial decompression in England.

► We provide international comparisons of the use of subacromial decompression surgery.

► Our data are from 2007/2008 onwards, so we underestimate the amount spent on subacromial decompression prior to publication of major clinical trial results (Can Shoulder Arthroscopy Work (CSAW) and Finnish Subacromial Impingement Arthroscopy Controlled Trial (FIMPACT)).

► There may be additional factors influencing surgery rates which we have not controlled for (eg, private health insurance coverage).

## INTRODUCTION

Health and social care services are 'straining at the seams' following increasing demand for services from an ageing population with more complex needs.[1] In England, over 200 clinical commissioning groups (CCGs) have a budget to purchase health services for their local populations.[2] Hospital care currently accounts for 48.5% (£74 billion) of government health expenditure in the UK.[3] It is vital that commissioners make evidence-based decisions to maximise the effectiveness of this hospital care budget to benefit the overall health of the population.

Medicines must be licensed for use for a particular condition, requiring pharmaceutical companies to provide evidence of effectiveness from clinical trials to relevant agencies such as the Medicines and Healthcare products

Regulatory Agency in the UK,[4] European Medicines Agency in the European Union or the Food and Drug Administration in the USA.[5] In the UK, the National Institute for Health and Care Excellence (NICE) also evaluates the cost-effectiveness of many medicines and does not recommend those which do not provide value. These regulatory processes have their limitations,[6] but require robust evidence for the introduction of new treatments. The quality of evidence required to introduce new surgical procedures is not as strict as for medicines,[4 5] in part because no specific product such as a drug or device is involved; it can be difficult to categorise procedures as 'new' rather than modifications and outcomes may depend on the skill of the practitioner as well as the procedure itself.[4] Once introduced, use of procedures can spread by clinical consensus,[5] and established practice and clinical evidence often take many years to be updated.[7 8]

National Health Service (NHS) England has recently commissioned a consultation regarding the use of 17 hospital procedures,[9] one of which is subacromial decompression for shoulder pain. Shoulder pain is common, with a lifetime prevalence of up to 66.7%.[10] Most of these cases (up to 70%) are related to rotator cuff tears or subacromial pain.[11] Subacromial pain is often considered to be caused by bony 'spurs' forming on the acromion, part of the shoulder blade, leading to inflammation in the surrounding bursa and tendons.[12 13] Subacromial decompression removes the bony spur on the acromion and releases the coracohumeral ligament.[13 14] There has been a rapidly increasing use of subacromial decompression in England, with over 21 000 procedures carried out in 2009/2010.[13]

Several randomised controlled trials (RCTs) since the early 1990s have compared subacromial decompression to non-operative treatment (eg, exercise) for shoulder pain and found no evidence of effectiveness.[15–17] Two recent multicentre RCTs including a placebo surgery arm have further questioned the effectiveness of subacromial decompression for shoulder pain.[18 19] The CSAW trial,[12 18] recruiting in England from 2012 to 2015, compared arthroscopic subacromial decompression surgery, placebo (investigational shoulder arthroscopy) and no treatment.[18] It found no difference in shoulder function after 6 months between the arthroscopic subacromial decompression group and the arthroscopy only (placebo) group, with a small, non-clinically significant benefit of surgery over the no treatment control. The FIMPACT trial,[19] recruiting in Finland from 2005 to 2013, compared subacromial decompression with placebo surgery and exercise therapy and echoed the results of CSAW, extending them to 2 years follow-up. A recent Cochrane review including CSAW, FIMPACT and earlier RCTs, found high-certainty evidence that subacromial decompression does not improve pain, function or health-related quality of life.[20] This seriously questions whether the resources invested in subacromial decompression represent good value for money for the NHS. As a result, a recent BMJ article made a strong recommendation against subacromial decompression surgery for chronic shoulder pain.[21]

In this study we use subacromial decompression for shoulder pain as an example to explore the relationship

| Any of these *diagnosis* codes in any position | | Any of these *procedure* codes in any position |
|---|---|---|
| M75.1 Rotator cuff syndrome | In combination with... | W84.8 Other specified therapeutic endoscopic operations on other joint structure |
| M75.3 Calcific tendinitis of shoulder | | Y52.8 Other specified approach to organ through other opening |
| M75.4 Impingement syndrome of shoulder | | Y76.7 Arthroscopic approach to joint |
| M75.5 Bursitis of shoulder | | W84.4 Endoscopic decompression of joint |

OR

| This *procedure* code in any position |
|---|
| O29.1 Subacromial decompression |

**Figure 1** ICD-10 and OPCS-4 codes used to define subacromial decompression.[13] ICD-10, International Classification of Diseases - version 10; OPCS-4, Office of Population, Censuses and Surveys- fourth revision.

between evolving evidence and clinical practice for hospital procedures, including how many procedures were performed over the last 10 years and how much money was spent before RCT evidence raised questions about the procedure's value; how procedure rates compare to other countries and how the NHS might reduce the numbers of these procedures.

## METHODS

### Data sources
Subacromial decompression procedures were identified using the 'admitted patient care' hospital episode statistics (HES-APC). HES is a routinely collected data set that records all episodes of care provided to patients admitted (day case or inpatient) to NHS hospitals in England and NHS-funded patients treated in the independent sector.[22 23] Each episode in HES represents a period of care under one consultant team. Up to 20 diagnoses are recorded per episode using the International Classification of Diseases (ICD) version 10. Up to 24 clinical procedures per episode may be recorded using Office of Population, Censuses and Surveys (OPCS) (fourth revision) codes. HES also includes the Lower Super Output Area of residence for each patient.[24]

### Identifying subacromial decompression
We extracted anonymised, individual episodes in the HES-APC (2007/2008 to 2016/2017) data set. We used diagnosis and procedure codes[13] (figure 1) to identify subacromial decompression. A small number of patients received multiple shoulder procedure episodes on the same day (0.3% of all episodes). When these were for the same procedure with the same laterality (0.25% of all episodes), we assumed coding error duplication so excluded the episodes. If a procedure was marked as bilateral (0.6%), this was counted as two procedures. We excluded patients who were not resident in England.

### Estimating procedure rates
National trends over time were estimated using directly standardised procedure rates[25] (per 100 000 population), with

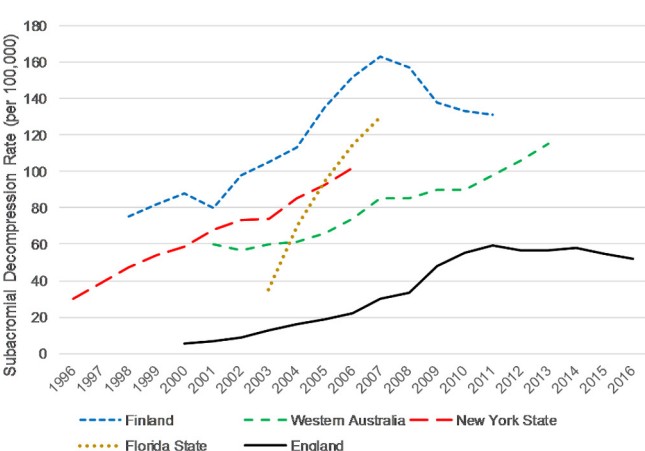

**Figure 2** Directly standardised rates (per 100 000 people) of subacromial decompression in England; Finland; New York State, USA; Florida State, USA and Western Australia notes for figure 2. England data prior to 2007 is taken from Judge *et al*.[13]; New York State data is for subacromial decompression with or without rotator cuff repair, while data for Florida State, Finland and Western Australia is for subacromial decompression alone.[37–40]

| | **2007/08** | **2016/17** |
|---|---|---|
| Table 1 Descriptive information for subacromial decompression patients, 2007/2008 and 2016/2017 | | |
| Procedure count | 15 112 | 28 802 |
| % Women | 51.0 | 52.0 |
| Age in years (SD) | 54.94 (12.55) | 54.89 (12.39) |
| % Arthroscopy | 39.0 | 94.1 |
| % Independent providers | 2.4 | 31.9 |
| % Day-case | 51.0 | 79.3 |

the population of England in 2016 as our standard population. For comparison of smaller areas, we estimated indirectly standardised rates[26] per 100 000 population, using the same standard population, and adjusting for deprivation and ethnicity (see online supplementary appendix A for more details).

### Estimating procedure costs

Costs were estimated for each financial year by linking Healthcare Resource Group codes for each admission in HES with the Department of Health Payment by Results National Tariffs for the appropriate financial year[27–36]; see online supplementary appendix A for more details.

### International comparisons

A search of MEDLINE and the Cumulative Index of Nursing and Allied Health (CINAHL) databases was conducted for the terms 'acromioplasty' or 'subacromial decompression' in conjunction with 'incidence' or 'prevalence' or 'epidemiology'. One author (TJ) screened the results for articles including rates of subacromial decompression contemporary with our data, and further screened cited articles within included studies, as well as articles which cited included studies.

All statistical analyses were conducted using Stata/MP 14.2 for Windows and we mapped variation in procedure rates across England in 2016/2017 using ArcGIS ArcMap 10.5.1 for Desktop.

### Patient and public involvement

There was no patient involvement in the design or conduct of this study. Two patients involved in the CSAW trial reviewed this manuscript; they were interested by the results and the cost-focused perspective.

## RESULTS

### The use of subacromial decompression in England

There were 15 112 subacromial decompression procedures (30 per 100 000 population) in 2007/2008, rising to 28 802 procedures (52 per 100 000 population) in 2016/2017 (figure 2), excluding those done in combination with rotator cuff repair. This represents a 91% increase in the number of subacromial decompressions over 10 years, with 266 692 procedures carried out in total. Most of this increase took place before 2011/2012, and procedure rates have slightly decreased between 2011/2012 and 2016/2017. While the gender balance and age of those having shoulder surgery have remained steady over the last decade, the proportion of procedures conducted as day cases, using arthroscopy, and/or by independent (ie, non-NHS) providers, have all increased (table 1).

### The cost of subacromial decompression in England

In 2016/2017, the median cost of an elective admission for subacromial decompression alone was £4476. The cost of subacromial decompression in England rose from £33 million in 2007/2008 to £125 million in 2016/2017. Over the 10-year period between 2007/2008 and 2016/2017 just under £1.1 billion was spent on subacromial decompression (excluding procedures done in combination with rotator cuff repair).

### Variation in use of subacromial decompression in England

In 2016/2017 there was substantial variation in procedure rates between CCGs, after adjusting for age, sex, deprivation and ethnicity profiles (figure 3). The map demonstrates pockets of very high use (>150% of the expected rate), for example in the Reading area, Wiltshire and East Lincolnshire. There were also areas where procedure rates were less than 50% of the expected rate, such as in Worcestershire, Gloucestershire, Swindon and North Norfolk. In 2016/2017 the ratio of procedure rates between a 'high use' CCG at the 90th percentile and a 'low use' CCG at the 10th percentile was 2.7 (95% CI: 2.2 to 3.4). This ratio is lower than the 2007/2008 ratio of 3.6 (95% CI: 2.2 to 6.1), although overlapping CIs suggest this may be due to chance variation; see table 2.

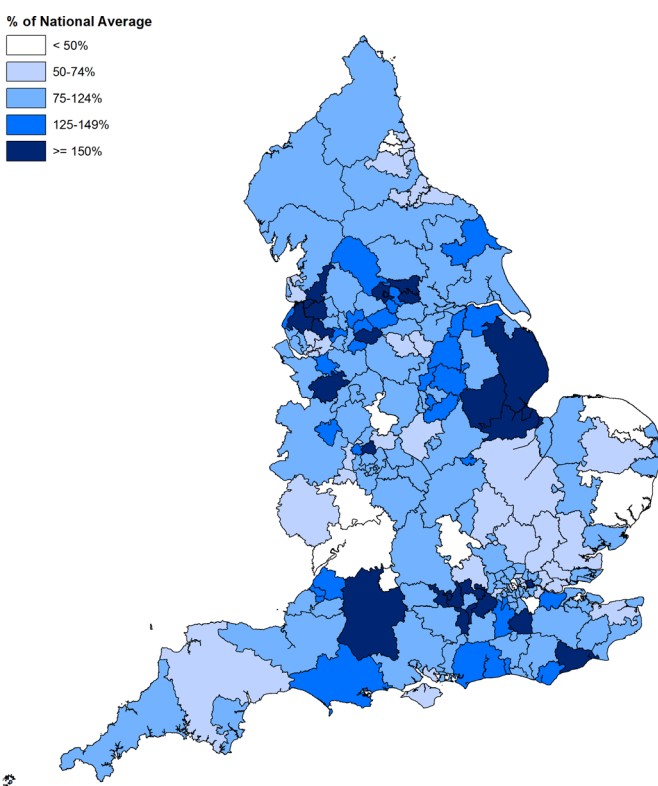

**% of National Average**

- ☐ < 50%
- ☐ 50-74%
- ☐ 75-124%
- ☐ 125-149%
- ☐ >= 150%

**Figure 3** Indirectly standardised rates of subacromial decompression by clinicalcommissioning group in England, 2016/2017.

### International comparison of rates of subacromial decompression

Table 3 shows rates of subacromial decompression in the most recent year available from England, Finland, Florida State, New York State and Western Australia. Rates in England were lower, often only half, that of other countries. For subacromial decompression alone, the procedure rates were lower in England (52 per 100 000 in 2016/2017) than Western Australia (roughly 115 per 100 000 in 2013),[37]

**Table 2** 90th/10th percentile ratios for directly age-sex standardised rates of subacromial decompression by clinicalcommissioning group, England, 2007/2008 to 2016/2017

| Year | 90th percentile | 10th percentile | 90th/10th ratio (95% CI*) |
|---|---|---|---|
| 2007/08 | 53 | 15 | 3.6 (2.2 to 6.1) |
| 2008/09 | 55 | 16 | 3.3 (2.1 to 5.2) |
| 2009/10 | 72 | 27 | 2.6 (2.0 to 3.5) |
| 2010/11 | 87 | 33 | 2.6 (1.9 to 3.6) |
| 2011/12 | 89 | 36 | 2.5 (2.0 to 3.1) |
| 2012/13 | 90 | 33 | 2.7 (2.0 to 3.7) |
| 2013/14 | 88 | 34 | 2.6 (2.1 to 3.3) |
| 2014/15 | 89 | 33 | 2.7 (2.0 to 3.7) |
| 2015/16 | 81 | 33 | 2.5 (1.4 to 4.3) |
| 2016/17 | 83 | 30 | 2.7 (2.2 to 3.4) |

*Confidence intervals for rate ratios.[61]

**Table 3** International comparisons of age-sex-standardised rates of subacromial decompression

| Article | Country | Data year | SAD rate (per 100 000 population) |
|---|---|---|---|
| Thorpe et al[37] | Western Australia | 2013 | ~115 |
| Paloneva et al[39] | Finland | 2011 | 131 |
| Vitale et al[40] | New York State | 2006 | 102 |
| Iyengar et al[38] | Florida State | 2007 | ~130 |
| Our Data | England | 2016/17 | 52 |

Numbers for Thorpe et al and Iyengar et al[38] were estimated from a graph; New York State data is for subacromial decompression with/without rotator cuff repair.
SAD, subacromial decompression.[37–40]

Florida State (130 per 100 000 in 2007)[38] and Finland (131 per 100 000 in 2011).[39]

Figure 2 compares trends in rates of subacromial decompression in England, Finland, Florida State, New York State and Western Australia. The rate of increase for subacromial decompression observed in our study (x2 between 2007/2008 and 2016/2017) was similar to Western Australia (x2 between 2001 and 2013),[37] Finland (x2.2 between 1998 and 2007)[39] and New York State (x2.5 between 1996 and 2006),[40] but lower than Florida State (x4.4 between 2003 and 2007).[38] The use of subacromial decompression in Finland peaked in 2007 and has since been declining, at least in publicly-funded hospitals, which has been attributed to accumulating evidence that it is no more clinically effective than non-surgical alternatives.[39]

## DISCUSSION

### Statement of principal findings

NHS England carries out nearly 30 000 subacromial decompression operations per year, at an annual cost of over £125 million. Between 2007/2008 and 2016/2017, 266 692 subacromial decompression procedures were carried out in England costing nearly £1.1 billion, before the addition of the CSAW and FIMPACT placebo-controlled trial results to the existing evidence prompted serious questions about the clinical benefit of the procedure. Rates of subacromial decompression alone in England have gradually declined since 2011/2012, although an increasing number are carried out in combination with rotator cuff repair. There was large variation between CCGs in England, even after adjustment for demographic variables, with 'high-use' areas carrying out nearly three times as many procedures as 'low-use' areas. Procedure rates in England were notably lower than other countries, arguing against any levelling of procedure rates being due to saturation of 'demand' for shoulder surgery.

## Strengths and weaknesses of the study

Our study used a national, longitudinal data set over a 10-year period covering all NHS secondary care providers in England. Hospital Episode Statistics are administrative rather than specifically designed for research. However, HES is also linked to payments for hospitals, which is a strong incentive to provide complete data, and allowed us to produce what we believe is the first exploration of costs associated with subacromial decompression in England. Payment by results tariffs are based on average national costs and may not reflect precise costs for each hospital admission. Population denominators, and linkage to the indices of multiple deprivation and census ethnicity data, allowed us to investigate trends and variations in procedure rates standardised on age, sex, deprivation and ethnicity. HES data records patients' area of residence, so we compared procedure rates based on place of residence rather than place of treatment. There may be other factors influencing rates which we have not controlled for (eg, private health insurance coverage). HES does not record procedures which are privately funded and provided, meaning our surgery rates are an underestimate of the population rate. We only provide cost data from 2007/2008 onwards, so we underestimate the amount spent on subacromial decompression prior to publication of the CSAW and FIMPACT trial results. International estimations of procedure rates do not use identical definitions of procedures and inclusion/exclusion criteria, but should be broadly comparable.

## Implications for policymakers and clinicians

NHS England spent over £1 billion on subacromial decompression during the last 10 years without having compelling evidence of clinical effectiveness or cost-effectiveness. Rates of subacromial decompression were already rising rapidly from 2000/2001 onwards.[13] It seems plausible that increasing awareness of concerns about the effectiveness of subacromial decompression surgery[15–17] and well-known recruitment to the CSAW trial tempered the rise in use of this surgery in England, otherwise more may have been spent. The CSAW trial involved 51 surgeons in 30 centres throughout the UK and was widely advertised and discussed among shoulder surgeons and shoulder physiotherapists. Extensive consultation was carried out by the trial team prior to and during the trial, including presentations at national meetings surveys and visits to individual surgeons and centres.[41] A similar plateau/decrease in procedures was observed in Finland after the commencement of the FIMPACT study in 2005 which involved only three centres in Finland (figure 2). It is likely that awareness of a potential lack of effectiveness of subacromial decompression had been growing in the years before CSAW and FIMPACT, based on earlier trial results.[15–17] However, it took well over a decade of increasing subacromial decompression use for clinical trial groups to run high quality, low risk-of-bias, placebo-controlled studies randomising a few hundred patients (313 patients for CSAW[18] and 210 in FIMPACT[19] to investigate its effectiveness. This delay may be due to perceived difficulties in recruiting patients to surgical trials with non-surgical comparators (eg,

UKUFF,[42] as well as known challenges of conducting surgical RCTs.[43] Methods to optimise recruitment, as used in CSAW and other trials,[44] are available to support the completion of such 'difficult' trials[45]; this should not now be a barrier to rapidly initiating trials to provide robust evidence about surgical interventions before they become widespread. More time is needed to see the longer-term impact of publication of the CSAW and FIMPACT results on subacromial decompression rates, both in the UK and internationally.

The NICE requires evidence of cost-effectiveness to recommend new medicines to be paid for by the NHS. It is unclear why the bar for introducing expensive surgical procedures should be significantly lower. A balance needs to be struck between supporting innovation in surgical procedures and preventing unnecessary treatment. New initiatives such as IDEAL (Idea, Development, Exploration, Assessment, Long-term Follow-up, Improving the Quality of Research in Surgery)[46] aim to provide such a regulatory framework for introducing new interventions.

It is important that new evidence is disseminated quickly without causing inequities in access to care. NICE published an updated Clinical Knowledge Summary for shoulder pain in April 2017[47] incorporating information from a commissioning guide published by the Royal College of Surgeons.[48] This recommended a range of conservative treatments from physiotherapy to corticosteroid injections, before surgery. However, many CCGs introduced their own criteria-based policies for access to shoulder surgery (eg, through Individual Funding Requests),[49] essentially meaning that commissioners would only pay providers for surgery under particular circumstances. These were implemented at different times and with different details, underlining the extent to which insufficient evidence may drive clinical and commissioner uncertainty,[50] and possibly leading to the wide variations shown across CCGs in our data. Where scientific evidence is applicable nationally or internationally, it would seem more efficient and appropriate to apply national policies to inform optimal use and encourage further research. There is a need to improve techniques for empirically-informed policy development in collaboration with relevant stakeholders.[51 52]

Despite the criticisms provided above, England has lower rates of shoulder surgery than other countries. The reasons for this are uncertain but could be due to differences in the health systems (eg, general practitioner gatekeeping of services), access to surgery and hospital reimbursement. Additionally, the National Institute of Health Research in England has funded major clinical trials on shoulder surgery,[18 42] as well as other procedures[53 54]; and is about to fund a further clinical trial to compare surgery with placebo surgery for partial thickness rotator cuff tears.[55] While the UK's national regulatory processes are imperfect, they may provide examples to learn from. However, these processes did not sufficiently constrain the use of subacromial decompression, a procedure later found to have little clinical benefit.

There have been several other controversies regarding the lack of effectiveness of procedures which have become

commonplace. One example is the use of stents to open narrowed arteries for treatment of stable angina (chest pain). Around half a million people receive stents for stable angina each year in the USA and Europe,[56] but a recent (RCT) including a placebo intervention found no difference in chest pain outcomes between inserting a stent and using standard medications.[57] Another example is arthroscopy to clean out the knee joint, on which around $4 billion is spent each year in the USA.[58] Recent RCTs,[59 60] including one using a placebo procedure as a comparison,[60] found no evidence of effectiveness to justify the spending. While we use subacromial decompression as an example in this study, our observations are likely to apply to interventional procedures more generally.

### Unanswered questions and future research

The example of subacromial decompression highlights that, in the absence of rigorous evaluation, costly interventions can proliferate over a long period of time. To maximise limited resources, it is vital that methods are developed to identify promising procedures early and commission trials to examine their value, as well as identify existing health technologies that may be ineffective, leading to overtreatment and wasting of resources.

There is an opportunity for a natural experiment exploring the impact of the results of the CSAW and FIMPACT trials[18 19] on the development of CCG policies, national guidelines and clinical decision-making with surgeons and patients. It is arguable that we should now see swift reductions in the use of subacromial decompression; research studies could help enhance the transfer of knowledge from trials into clinical practice.

### CONCLUSIONS

NHS England pays for nearly 30 000 shoulder subacromial decompression procedures each year at an annual cost of over £125 million, with little evidence that they are effective or cost-effective. The rates of this operation in other countries are even higher. This raises serious questions around the regulatory and professional processes governing the adoption and widespread use of surgical interventions. High quality RCTs should be funded early to examine the effectiveness and cost-effectiveness of expensive procedures using methods to optimise recruitment, and robust processes should be developed to reduce the use of ineffective procedures.

**Acknowledgements** The authors would like to thank two patients from the CSAW trial, Carol Brennan and Dair Farrer-Hockley, who took the time to read and comment on this manuscript.

**Contributors** This publication is the work of the authors, who serve as guarantors for the contents of this paper. TJ contributed to study design, data cleaning, data analysis, interpretation of results and writing the manuscript. M-JL contributed to study design, data cleaning, interpretation of results and writing the manuscript. AJC contributed to study design, interpretation of results and writing the manuscript. DB, LR, and JD contributed to interpretation of results and writing the manuscript. WH contributed to study conceptualisation, study design, interpretation of results and writing the manuscript. TJ had full access to the data in the study

and takes responsibility for the integrity of the data and the accuracy of the data analysis.

**Funding** This research was funded by the National Institute for Health Research (NIHR) Collaboration for Leadership in Applied Health Research and Care West (NIHR CLAHRC West). The views expressed in this article are those of the author(s) and not necessarily those of the NHS, the NIHR or the Department of Health and Social Care.

**Map disclaimer** The depiction of boundaries on the map(s) in this article do not imply the expression of any opinion whatsoever on the part of BMJ (or any member of its group) concerning the legal status of any country, territory, jurisdiction or area or of its authorities. The map(s) are provided without any warranty of any kind, either express or implied.

**Competing interests** All authors have completed the ICMJE uniform disclosure form at www.icmje.org/ coi_disclosure.pdf and declare: TJ and JD had financial support from NIHR CLAHRC West for the submitted work; no financial relationships with any organisations that might have an interest in the submitted work in the previous three years; no other relationships or activities that could appear to have influenced the submitted work.

**Patient consent for publication** Not required.

**Ethics approval** We were provided with routinely-collected Hospital Episode Statistics data under licence from NHS Digital (DARS-NIC-17875-X7K1V). The licence allows us to use the information under Section 261 of the Health and Social Care Act 2012, 2(b)(ii): "after taking into account the public interest as well as the interests of the relevant person, considers that it is appropriate for the information to be disseminated".

**Provenance and peer review** Not commissioned; externally peer reviewed.

**Data availability statement** Data may be obtained from a third party and are not publicly available.

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
