## [Reviewer comments · BMJ Open]

ARTICLE DETAILS

TITLE (PROVISIONAL)	A longitudinal study of use and cost of subacromial decompression surgery: the need for effective evaluation of surgical procedures to prevent overtreatment and wasted resources.
AUTHORS	Jones, Tim; Carr, Andrew; Beard, David; Linton, Myles-Jay; Rooshenas, Leila; Donovan, Jenny; Hollingworth, William

VERSION 1 – REVIEW

REVIEWER	Tuomas Lähdeoja Helsinki University Hospital and FICEBO Helsinki Finland
REVIEW RETURNED	10-Apr-2019

GENERAL COMMENTS	Reviewer comments I thank for the opportunity to review this manuscript, which addresses an important issue in a clear and logical way. The efforts to change ineffective practices are sorely needed across the world. I have two major issues and some more minor comments and suggestions. The major issues: 1: Even before the CSAW and FIMPACT trials, there were a number of trials (Brox 1993, Haahr 2005, Ketola 2009 and Peters 1997) investigating the effectiveness of SAD, with uniform results of no benefit. I think that these should be mentioned across the text – now it reads like there was no evidence whatsoever before the new high-quality trials. In my mind it is even more amazing that the number of surgeries rose so high, with well-conducted, if open label, RCTs NOT supporting the procedure, and it took two placebo-controlled trials to raise the attention to the issue, something which might be mentioned in the discussion. 2: Subacromial decompressions with rotator cuff repair are included in Fig2 and in the text as well. I believe including the SADs done in conjunction with RC repair dilutes the message of the article. Leaving those procedures out altogether would make the manuscript clearer, since SAD in conjunction with RC repair is a minor addition to the RC repair procedure, and can sometimes be done to make the RC repair technically easier, whereas sole SAD is the only reason for the surgery and associated treatment burden and cost. True, there is evidence that adding SAD to RC repair does not affect the outcome and is generally not necessary, but I
---

	think it is very different than a patient undergoing acromioplasty as the sole procedure. Concentrating on the acromioplasty only surgeries would keep the message of this paper (loud and) clear. Also the poor effectiveness of RC repairs when looking at the RCT evidence is another issue, but not in the scope of this paper. Minor issues: Line 162 – how many patients? 185-188 + appendix A – I can't really appraise the trustworthiness of the numbers and the methods, operating outside the English system. 197 – the confidence intervals overlap – can you say this is a true decrease? 209 – See major issue 2, but if the SADs during RC repairs are kept, should the rates of SAD with RC repair be presented earlier? 268-269 – The discussion in the (shoulder) surgical / orthopaedic community in Finland was not strongly driven by the ongoing FIMPACT trial at that time, but more by some individuals (some of them involved in the Fimpat trial, some in the Ketola 2009 trial, but also surgeons not participating in the trials) raising concerns about the effectiveness of SAD based on the earlier RCTs, especially Brox 1993 and Haahr 2005, and the lack of any convincing evidence to support the use of the procedure. 300-302 – The possibility of a subgroup who might benefit from SAD is not supported by the current evidence, and there is no evidence on who these patients might be. I suggest deleting the sentence “It is also important to note that certain patients may still benefit from surgery.” and strongly consider deleting the following sentence too. 258 the whole chapter Implications for policymakers and clinicians – maybe consider adding something about the funder “decommissioning” the procedure by direct action by moving to not paying for the surgeries. An example of this can be found in Finland where the Council for Choices in Health Care in Finland (COHERE Finland) decided in Feb 2017 that arthroscopy degenerative knee conditions is not part of the publicly funded healthcare repertoire: https://palveluvalikoima.fi/documents/1237350/4120541/Recommendation+-+Knee+degeneration+treatment+by+keyhole+surgery/48d6e248-a81f-493a-8c4a-f946e8ae308b/Recommendation+-+Knee+degeneration+treatment+by+keyhole+surgery.pdf I understand that in Canada similar action has been taken by changing the reimbursement (paid by the public funder for the hospital) price for an arthroscopy for degenerative knee conditions to very low, which has effectively discouraged the widespread use of the operation.
--	--

VERSION 1 – AUTHOR RESPONSE

Tuomas Lähdeoja (Reviewer 1)	
Even before the CSAW and FIMPACT trials, there were a number of trials (Brox 1993, Haahr 2005, Ketola 2009 and Peters 1997) investigating the effectiveness of SAD, with uniform results of no benefit. I think that these should be mentioned across the text – now it reads like there was no evidence whatsoever before the new high-quality trials. In my mind it is even more amazing that the number of surgeries rose so high, with well-conducted, if open label, RCTs NOT supporting the procedure, and it took two placebo-controlled trials to raise the attention to the issue, something which might be mentioned in the discussion.	We thank the reviewer for this suggestion and have updated the introduction (lines 104-120) and the ‘Statement of Principle Findings’ and ‘Implications for policy makers’ sections in the discussion to include these earlier RCTs, as well as a recent Cochrane review (2019) which includes all of the relevant studies.
Subacromial decompressions with rotator cuff repair are included in Fig2 and in the text as well. I believe including the SADs done in conjunction with RC repair dilutes the message of the article. Leaving those procedures out altogether would make the manuscript clearer, since SAD in conjunction with RC repair is a minor addition to the RC repair procedure, and can sometimes be done to make the RC repair technically easier, whereas sole SAD is the only reason for the surgery and associated treatment burden and cost. True, there is evidence that adding SAD to RC repair does not affect the outcome and is generally not necessary, but I think it is very different than a patient undergoing acromioplasty as the sole procedure. Concentrating on the acromioplasty only surgeries would keep the message of this paper (loud and) clear. Also the poor effectiveness of RC repairs when looking at the RCT evidence is another issue, but not in the scope of this paper.	We agree that it improves the clarity of the paper to focus on subacromial decompression alone. We have removed the line on Figure 2 for England including rotator cuff repair, updated Table 3, and references to this in the text (lines 176-177, 209-211, 282-284).
Line 162 – how many patients?	We have updated line 162 to clarify that two patients from the CSAW trial reviewed the manuscript.
185-188 + appendix A – I can’t really appraise the trustworthiness of the numbers and the methods, operating outside the English system.	The Grasic et al. (2015) reference in Appendix A outlines the payment process for English hospitals which we have tried to follow

197 – the confidence intervals overlap – can you say this is a true decrease?	We have updated lines 196-197 to mention the overlapping confidence intervals and that the decrease in the ratio may be due to chance
209 – See major issue 2, but if the SADs during RC repairs are kept, should the rates of SAD with RC repair be presented earlier?	As recommended, we have removed references to SAD with RC to clarify our message, including line 209.
268-269 – The discussion in the (shoulder) surgical / orthopaedic community in Finland was not strongly driven by the ongoing FIMPACT trial at that time, but more by some individuals (some of them involved in the Fimpat trial, some in the Ketola 2009 trial, but also surgeons not participating in the trials) raising concerns about the effectiveness of SAD based on the earlier RCTs, especially Brox 1993 and Haahr 2005, and the lack of any convincing evidence to support the use of the procedure.	We have updated the ‘implications for policy makers’ section in the discussion to mention clinician awareness of the earlier RCTs.
300-302 – The possibility of a subgroup who might benefit from SAD is not supported by the current evidence, and there is no evidence on who these patients might be. I suggest deleting the sentence “It is also important to note that certain patients may still benefit from surgery.” And strongly consider deleting the following sentence too.	We have removed these lines as recommended.
258 the whole chapter Implications for policymakers and clinicians – maybe consider adding something about the funder “decommissioning” the procedure by direct action by moving to not paying for the surgeries. An example of this can be found in Finland where the Council for Choices in Health Care in Finland (COHERE Finland) decided in Feb 2017 that arthroscopy degenerative knee conditions is not part of the publicly funded healthcare repertoire: https://palveluvalikoima.fi/documents/1237350/4120541/Recommendation+-+Knee+degeneration+treatment+by+keyhole+surgery/48d6e248-a81f-493a-8c4af946e8ae308b/Recommendation+-+Knee+degeneration+treatment+by+keyhole+surgery.pdf	We have added a line (around line 295) in the ‘implications for policy makers’ section of the discussion to make clear that this withholding of payment by commissioners is already a tool in use, but that it should be done at a national level where evidence is clear, instead of each area adopting different rules.

I understand that in Canada similar action has been taken by changing the reimbursement (paid by the public funder for the hospital) price for an arthroscopy for degenerative knee conditions to very low, which has effectively discouraged the widespread use of the operation.	
---	--

VERSION 2 – REVIEW

REVIEWER	Tuomas Lähdeoja Helsinki University Hospital Finland
REVIEW RETURNED	13-Jul-2019

GENERAL COMMENTS	My earlier concerns/comments have been addressed commendably. A very good article. One point, though: The paper might benefit from a more "catchy" (and shorter) title, maybe discuss with the editor? Something along the lines (though this is probably too tabloid) "Billion wasted in shoulder surgery: to prevent overtreatment and waste of resources, surgical procedures need effective evaluation before widespread adoption."
---